# Development and Validation of a Nomogram Based on Metabolic Risk Score for Assessing Lymphovascular Space Invasion in Patients with Endometrial Cancer

**DOI:** 10.3390/ijerph192315654

**Published:** 2022-11-25

**Authors:** Jingyuan Wang, Xingchen Li, Xiao Yang, Jianliu Wang

**Affiliations:** Department of Obstetrics and Gynecology, Peking University People’s Hospital, Beijing 100044, China

**Keywords:** metabolism, lymphovascular space invasion, endometrial cancer, nomogram

## Abstract

Objective: This study assessed the predictive value of the metabolic risk score (MRS) for lymphovascular space invasion (LVSI) in endometrial cancer (EC) patients. Methods: We included 1076 patients who were diagnosed with EC between January 2006 and December 2020 in Peking University People’s Hospital. All patients were randomly divided into the training and validation cohorts in a ratio of 2:1. Data on clinicopathological indicators were collected. Univariable and multivariable logistic regression analysis was used to define candidate factors for LVSI. A backward stepwise selection was then used to select variables for inclusion in a nomogram. The performance of the nomogram was evaluated by discrimination, calibration, and clinical usefulness. Results: Independent predictors of LVSI included differentiation grades (G2: OR = 1.800, 95% CI: 1.050–3.070, *p* = 0.032) (G3: OR = 3.49, 95% CI: 1.870–6.520, *p* < 0.001), histology (OR = 2.723, 95% CI: 1.370–5.415, *p* = 0.004), MI (OR = 4.286, 95% CI: 2.663–6.896, *p* < 0.001), and MRS (OR = 1.124, 95% CI: 1.067–1.185, *p* < 0.001) in the training cohort. A nomogram was established to predict a patient’s probability of developing LVSI based on these factors. The ROC curve analysis showed that an MRS-based nomogram significantly improved the efficiency of diagnosing LVSI compared with the nomogram based on clinicopathological factors (*p* = 0.0376 and *p* = 0.0386 in the training and validation cohort, respectively). Subsequently, the calibration plot showed a favorable consistency in both groups. Moreover, we conducted a decision curve analysis, showing the great clinical benefit obtained from the application of our nomogram. However, our study faced several limitations. Further external validation and a larger sample size are needed in future studies. Conclusion: MRS-based nomograms are useful for predicting LVSI in patients with EC and may facilitate better clinical decision-making.

## 1. Introduction

Endometrial cancer (EC) is the most common gynecological malignancy in the United States and its related mortality is on the rise [1]. Similar results have been observed in developing countries, with an increasing incidence and mortality [2,3]. Symptoms such as abnormal vaginal bleeding occur early in most women; thus, many patients are diagnosed at an early stage with a 5-year disease-specific survival rate of 80–85% [4]. The standard primary treatment for EC is surgery, consisting of a total hysterectomy and bilateral salpingo-oophorectomy with a lymph node assessment. Although the benefits of lymphadenectomy remain debated, it is recommended to perform a lymphadenectomy when lymphovascular space invasion (LVSI) is present in the European Society of Gynecological Oncology (ESGO) guidelines [5]. LVSI is defined as the presence of malignant tumor cells in the lymphatic and/or vascular vessels and identified in up to 34% of EC [6,7]. Although LVSI has no influence on the 2009 International Federation of Gynecology and Obstetrics (FIGO) stage of surgery, it has been widely recognized as an important factor leading to poor outcomes such as lymph node metastasis, and lower progression-free and overall survival rates [8,9,10]. As LVSI is an independent risk factor of lymph node metastasis (LNM), preoperative understanding of the status of LVSI may assist clinicians with optimizing a treatment plan, such as the adequacy of resection of para-aortic lymph nodes in patients with LVSI-positive tumors [11]. However, LVSI can currently only be determined after surgical resection of the uterine corpus.

Therefore, a preoperative and noninvasive prediction model is needed to evaluate the probability of LVSI. Currently, nomograms have been widely considered useful and accessible tools to estimate the risk score of each patient by incorporating potential clinical and pathological characteristics for oncologic outcomes, as has been widely demonstrated in numerous types of cancers [12,13]. Moreover, there are several established nomograms for screening lymph node metastasis (LNM), ovarian metastasis, coexisting adnexa malignancy, recurrence, and overall and cancer-specific survival in EC patients [14,15,16,17]. However, to our knowledge, correct prognostic prediction nomograms for LVSI are limited.

Recent years have shown an increased interest in the association between malignant tumors and metabolic syndrome (MetS): a cluster of obesity, insulin resistance, hypertension, dyslipidemia, and other metabolic abnormalities [18,19]. Several studies have shown that EC is one of the cancers most strongly associated with metabolic diseases and the incidence of EC is increasing with the rise in the prevalence of metabolic diseases [20,21]. Metabolic parameters can be obtained in a non-invasive and cost-effective way. Recently, a metabolic risk score (MRS) based on a panel of markers including body mass index (BMI), pulse pressure (PP), fasting plasma glucose (FPG), triglyceride (TG), and high density lipoprotein cholesterol (HDL-C) has been considered a better index to provide information on metabolic status, as its superior ability to predict has been confirmed compared with the model containing traditional clinicopathologic characteristics in a variety of tumors, such as esophageal cancer [22]. We adopted the MRS constructed by the Institute of Clinical Medical Sciences, China-Japan Friendship Hospital in Beijing, China when all five metabolism-related factors were analyzed in quintiles. The detailed process is shown in Table 1. However, the relationship between the MRS and LVSI in EC patients has not yet been reported. Therefore, this study establishes an MRS-based nomogram for the preoperative prediction of LVSI in EC patients to help with decision-making in clinical practice.

## 2. Materials and Methods

Patient selection 

We retrospectively reviewed patients who were diagnosed with EC between January 2006 and December 2020 at the department of obstetrics and gynecology in Peking University People’s Hospital. Patients whose pathology was confirmed as endometrial cancer by histology were eligible for inclusion. Other inclusion criteria included patients older than 18 years. The exclusion criteria were as follows: (1) combination with other malignant tumors; (2) absence of medical records; (3) a history of any preoperative therapy. After strict inclusion and exclusion criteria, a total of 1076 cases were included for further analysis. This retrospective study was approved by the ethics committee board of Peking University People’s Hospital and followed the Declaration of Helsinki.

Cohort definition and variable recode 

After enrollment, all patients were randomly divided into the training and validation cohorts in a ratio of 2:1 using a random number method in SPSS statistical software (version 24.0). The training cohort of 718 patients was used to screen variables and construct the nomogram, and the validation cohort of 358 patients was used as a validation data set for internal validation. According to a previous literature review, the following variables were retrieved for each patient from the electronic medical records. The variables included:(1)basic demographics, including age at diagnosis, menopause status, BMI;(2)comorbidities, including diabetes, hypertension;(3)vital signs, including systolic blood pressure (SBP), diastolic blood pressure (DBP), PP;(4)laboratory tests, including FPG, total cholesterol (TC), TG, HDL-C, MRS, cancer antigen 125 (CA125), cancer antigen 199 (CA199);(5)pathological features including LVSI, ascites cytology, histology, grade, LNM, myometrial invasion (MI), cervical stromal invasion (CI).

Statistical analysis 

Categorical variables were expressed as frequency and percentage (%) and as mean ± standard deviation (SD) for continuous variables. To compare differences between the training and validation cohorts, we used two-sample *t*-tests for normally distributed continuous variables and Wilcoxon rank-sum tests for non-normally distributed continuous variables. Chi-square or Fisher’s exact tests were applied for categorical variables. Univariable logistic regression analysis was used to evaluate the associations of the risk of LVSI in patients suffering from EC within all clinicopathological parameters. All variables that in univariate analysis showed significant differences with *p* values < 0.05 were entered into a multivariable logistic analysis. A backward stepwise selection was then used to select candidate variables for inclusion in a nomogram. Then, a nomogram model for LVSI probability was developed.

The performance of the nomogram was evaluated by discrimination, calibration, and clinical usefulness. First, the area under the receiver operating characteristic curve (AUC) was calculated to quantify the power of discrimination. Values ranged from 0.5 to 1.0, where 0.5 represents no discrimination and 1.0 indicates complete discrimination. Typically, a value greater than 0.7 suggests a reasonable estimation. Next, the calibration of the nomogram was determined by a calibration plot to graphically visualize how far the predictions were from the actual observation, displaying mean nomogram-based predictions on the horizontal axis versus the actual observed LVSI probabilities in both the training and the validation cohorts. Finally, decision curve analysis (DCA) was used to assess clinical effectiveness by quantifying the standardized net benefits at different threshold probabilities. Such analyses can determine the ability of a model to predict fine-scale outcomes based on a set of risk parameters.

Statistical analyses were conducted by using the statistical software package R (http://www.R-project.org The R Foundation, accessed on 1 June 2021) and Empower-Stats (http://www.empowerstats.com, accessed on 1 June 2021, X&Y Solutions, Inc, Boston, MA, USA). Unless otherwise indicated, all tests were two-sided and the *p* value < 0.05 was considered as statistically significant.

## 3. Results

### 3.1. Patient Characteristics

Based on the inclusion criteria and exclusion criteria, a total of 1076 patients with EC from 2006 to 2020 in Peking University People’s Hospital were enrolled in this study. In addition, these eligible patients were randomly divided into the training cohort (*n* = 718) and the internal validation cohort (*n* = 358) in a ratio of 2:1. In the training cohort, the majority of the patients were ≤60 years old (68.11%) and the mean MRS was 2.36 ± 4.31. Among these patients, most patients (92.06%) were negative for LNM and as far as tumor grade, 265 (36.91%) patients had a low-risk tumor grade, 324 (45.13%) patients had a moderate-risk tumor grade, and 129 (17.97%) had a high-risk tumor grade. In the validation cohort, patients younger than 60 years old accounted for 65.92% and the mean MRS was 2.58 ± 4.31. Among these patients, most patients (90.50%) were negative for LNM and as far as tumor grade, 133 (37.15%) patients had a low-risk tumor grade, 152 (42.46%) patients had a moderate-risk tumor grade, and 73 (20.39%) had a high-risk tumor grade. Additionally, the overall incidence of LVSI was 16.85% (121/718) in the training cohort and 16.76% (60/358) in the validation cohort, respectively. The rest of the results of demographics and clinicopathological characteristics of patients in the training and validation cohorts are shown in Table 2. There were no significant differences regarding all the enrolled variables between the two cohorts (all *p* > 0.05).

### 3.2. Risk Factors of LVSI

To identify potential risk factors of LVSI occurrence, univariable logistic regression analysis was performed in EC patients (Table 3). Univariable logistic regression indicated that seven candidate factors namely, age, MRS, histology, differentiation grade, CA125, menopause status, and MI were positively associated with LVSI development (*p* < 0.05). Subsequently, the multivariable logistic regression analysis was utilized to define the independent prognostic markers and estimate their effect on the presence of LVSI for patients in the training cohort. Significant independent predictors of LVSI included differentiation grade (G2: OR = 1.800, 95% CI: 1.050–3.070, *p* = 0.032) (G3: OR = 3.49, 95% CI: 1.870–6.520, *p* < 0.001), histology (OR = 2.723, 95% CI: 1.370–5.415, *p* = 0.004), MI (OR = 4.286, 95% CI: 2.663–6.896, *p* < 0.001), and MRS (OR = 1.124, 95% CI: 1.067–1.185, *p* < 0.001). The detailed results of the multivariate logistic regression analysis are presented in Table 3. The analysis of the validation group showed similar results as shown in Table 4.

### 3.3. Nomogram Development and Validation

A nomogram was established to predict a patient’s probability of developing LVSI based on the four independent risk factors above: histology, differentiation grade (I, II, or III), MRS, and MI (Figure 1). Among them, MI was evaluated using imaging techniques, such as ultrasound or magnetic resonance imaging (MRI). From the nomogram, MRS had the greatest influence on LVSI, followed by MI, histology, and differentiation grade. Lines are drawn straight upward to the points axis to assign a weighted score to each of the independent risk factors. The number of points received for each variable value on the point scale are added together. The total points reflect the sum of the score of each factor, and are then converted to a probability of LVSI for a given patient by drawing lines straight down from the total points axis to the risk of LVSI axis. The highest total score was 220 points, and the scale of the LVSI probability ranged from 0.1 to 0.9. Therefore, a larger total point score indicated the greater possibility of LVSI.

The performance of the nomogram was evaluated by discrimination, calibration, and clinical usefulness. First of all, we examined the discrimination of the nomogram, which demonstrated high discrimination and good prognostic accuracy as indicated by the AUC value of 0.790 (95% CI 0.7438–0.8357) for the MRS-based nomogram and 0.758 (95% CI 0.7110–0.8055) for the clinicopathologic nomogram, with a significant *p* value (*p* value = 0.0376) in the training cohort as shown in Figure 2. In the validation cohort, the AUC value was 0.769 (95% CI 0.7018–0.8360) for the MRS-based nomogram and 0.709 (95% CI 0.6367–0.7809) for the clinicopathologic nomogram with a significant *p* value (*p* value = 0.0386). Subsequently, the calibration plot showed a favorable consistency between the predicted and actual probability in both the training and the validation groups as shown in Figure 3, which indicates adequate fit of the nomogram for predicting LVSI. To further evaluate the clinical benefit of nomogram performance, we conducted the DCA, showing the great benefit obtained from the application of our nomogram, as shown in Figure 4.

## 4. Discussion

In this study, we developed and validated a nomogram for predicting LVSI in patients with EC, including model 1 (traditional clinicopathological signature) and model 2 (ana MRS was added to model 1 to form model 2). Our results indicate that the predictive value of model 2 for LVSI is higher than that of model 1, identifying that the addition of an MRS can improve the predictive ability of LVSI in patients with EC.

LVSI has been reported to be an adverse prognostic factor for EC and it has been shown to indicate a significant difference in overall survival rates between patients with and without LVSI [23]. Several studies have established the association between the presence of LVSI and multiple negative prognostic factors, including a higher incidence of lymph node spread [11,24]. The occurrence of LVSI associates a 7.9 relative risk of nodal metastases [11,25]. Furthermore, in a study published by Jorge et al., the estimated risk of lymphatic dissemination was 21% and 2.1% in positive and negative LVSI groups respectively [9]. It has been reported in previous literatures that lymphadenectomy is related to increased complications with no survival benefit [26,27]. Hence, a more tailored and individualized lymphadenectomy is needed to avoid overtreatment. As the first stage of lymphatic dissemination [28], the status of LVSI should be determined whenever a decision to perform or omit lymphadenectomy is made [29,30].

According to the European Society of Medical Oncology, the European Society of Gynecological Oncology, and the European Society of Radiotherapy and Oncology guidelines (ESMO-ESGO-ESTRO), patients with nonendometrioid histology, poorly differentiation, deep stromal invasion, and LVSI are considered to be at higher risk and recommended to receive adjuvant treatment [31]. Many noninvasive methods have been used preoperatively to evaluate adverse factors in patients at high risk. The histology and differentiation of the tumor can be assessed by hysteroscopy. For the detection of deep stromal invasion, the sensitivity and specificity in conventional ultrasound is comparable to MRI [32]. While there are feasible and accurate screening tools to identify those risk factors, few studies have concentrated on the preoperative prediction of the status of LVSI, making LVSI the bottleneck in selecting appropriate treatment. Hence, it is of great importance in clinical practice to predict LVSI before surgery. In the current study, we are the first to establish a nomogram based on an MRS to predict LVSI outcomes in EC patients.

A total of 1076 patients with EC from 2006 to 2020 in Peking University People’s Hospital were enrolled in this study to develop and validate a nomogram for clinicians to predict the presence of LVSI in patients with EC and conduct tailored treatment. All patients were randomly divided into the training and validation cohorts in a ratio of 2:1. Seven candidate factors were indicated by univariable logistic regression, including age, MRS, histology, differentiation grade, CA125, menopause status, and MI and were positively associated with LVSI development. Further multivariable logistic regression analysis was utilized to define the four independent prognostic markers, namely MI, histology, MRS, and differentiation grade. A nomogram was then established to predict a patient’s probability of developing LVSI based on the four independent risk factors above. We then compared the performance of the model combining an MRS with that of a single model based on MI, histology, and differentiation grade in predicting LVSI in patients with EC. Consequently, the model based on the MRS achieved encouraging performance in both the training and validation cohorts according to the AUC (AUC value = 0.790, 95% CI 0.7438–0.8357 in the training cohort and AUC value = 0.769, 95% CI 0.7018–0.8360 in the internal validation cohort), suggesting that the model based on the MRS is superior to the single clinicopathologic model in predicting LVSI. Subsequently, the calibration plot showed a favorable consistency in both the training and the validation groups. DCA showed that the use of the combined nomogram with an MRS could add more benefit compared with the single clinicopathologic model.

Previous literature has reported several risk-scoring models for predicting LVSI in patients with EC. Yan Luo et al. developed a multiparametric MRI-based radiomics nomogram to assess LVSI preoperatively for providing decision-making support [33]. In Ling Long et al.’s study, the computer-vision signature was added to construct a nomogram, which showed better predictive ability compared with the model based on traditional handcrafted radiomics features [34]. It was reported that single pathological characteristics, including tumor diameter, depth of myometrial invasion, tumor grade, and cervical stromal involvement, were incorporated in an innovative score called the LVSI index to determine LVSI [35]. A recent study carried out by Vito Andrea Capozzi et al. also introduced a novel preoperative predictive score to estimate the status of LVSI, including features of tumor grade, serum CA125 levels, myometrial invasion, and tumor size [36]. It has also been reported that CA125 ≥ 21.2U/mL and fibrinogen ≥ 2.58mg/dL were valuable for predicting LVSI in EC women [37]. Some researchers also concentrated on exploring candidate genes to determine the potential predictors of LVSI, suggesting the signature of 55-gene contributed to predicting the presence of LVSI in EC [23].

In this study, we developed a novel prognostic nomogram based on an MRS for predicting LVSI in EC for the first time. Convincing evidence has appeared to support a close connection between MetS and tumors [38,39,40]. A panel of metabolic factors are involved both in tumorigenesis and its progression and MetS was associated with poor prognosis in EC patients [41,42]. Currently, there is no consensus on the best definition of MetS, but based on the criteria suggested by the Chinese Diabetes Society in 2004, metabolic syndrome represents a cluster of three or more cardiovascular risk factors, including obesity, hyperglycemia, hypertension, high circulating TG, and low circulating high-density lipoprotein cholesterol (HDLC). In Hong Sha et al.’s study, an MRS grading system adopting five metabolism-related markers was first described and the question was raised of whether a nomogram based on an MRS was superior when it came to predicting postsurgical esophageal cancer-specific mortality over traditional clinicopathologic parameters [22]. Similarly, in our study, an MRS was also added to construct the nomogram for predicting LVSI in EC patients, showing superior predictive performance and clinical applicability.

To our knowledge, the relationship between metabolic disease and LVSI has not been reported in literature and our study is the first to identify and validate the MRS-based nomogram to predict LVSI in EC patients. Moreover, the parameters of the nomogram are available non-invasively and cost-effectively. One of the other strengths of our study lies in the fact it uses the largest sample size (*n* = 1076) to date.

However, our study faced several limitations. Firstly, all the data were derived from the same institution and the current study lacks any external validation procedure. Hence, further external validation is needed to validate this nomogram. Secondly, although the number of enrolled cases is the largest in published literature when predicting LVSI in EC patients, a larger sample size is needed in future studies.

## 5. Conclusions

An MRS-based nomogram is useful for predicting LVSI in patients with EC and may facilitate better clinical decision-making.

## Figures and Tables

**Figure 1 ijerph-19-15654-f001:**
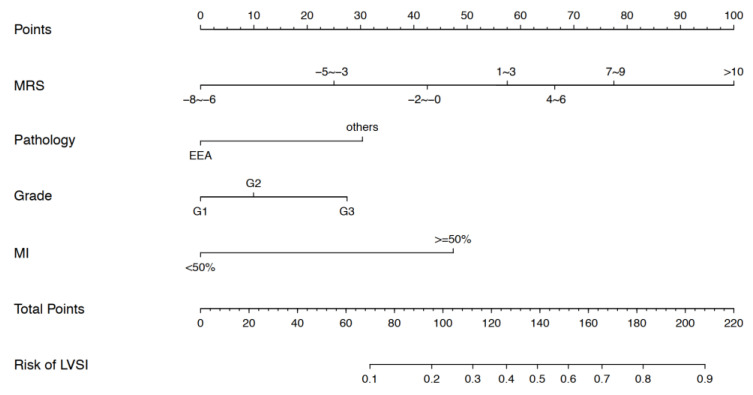
Nomogram to predict LVSI for EC patients.

**Figure 2 ijerph-19-15654-f002:**
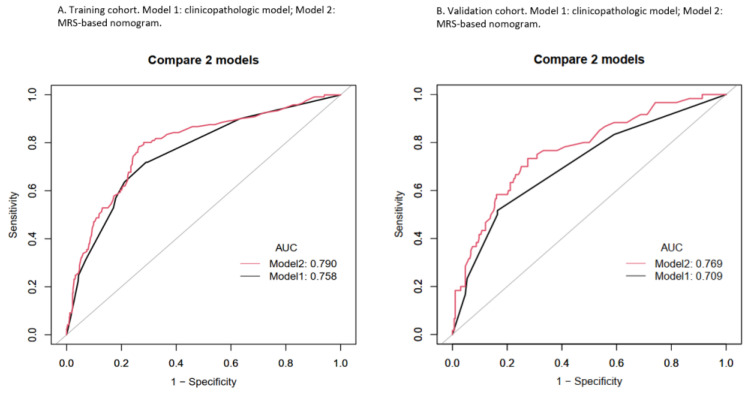
AUC of clinicopathologic model and MRS-based nomogram for prediction of LVSI in EC in (**A**) training cohort and (**B**) validation cohort.

**Figure 3 ijerph-19-15654-f003:**
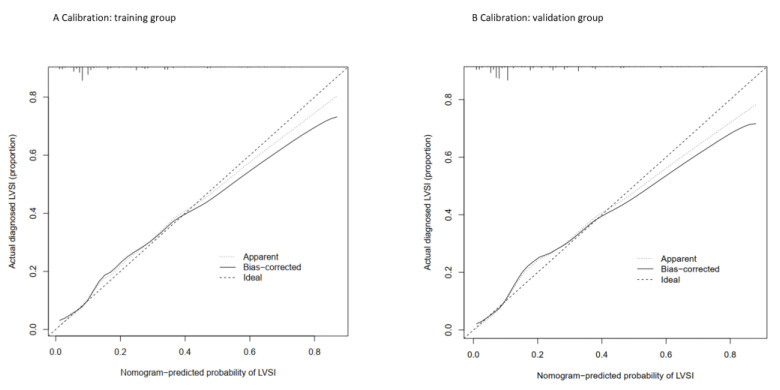
Calibration plots of the nomogram to predict LVSI in (**A**) training cohort and (**B**) validation cohort.

**Figure 4 ijerph-19-15654-f004:**
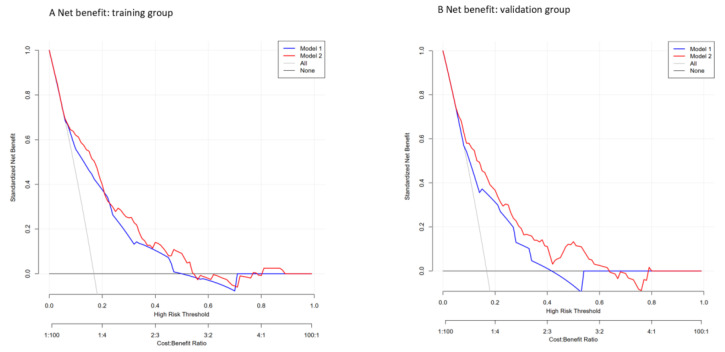
DCA of the clinicopathologic and MRS-based nomogram for predicting LVSI in the training cohort and validation cohort. Solid blue line = net benefit when all EC patients are considered as not having the outcome (negative LVSI); red dashed line = net benefit when all EC patients are considered as having the outcome (positive LVSI).

**Table 1 ijerph-19-15654-t001:** The detailed information about calculation of MRS.

Metabolic Risk Factors	Quintile: Range	Score
BMI	Q1: 17.15–19.72	−1
	Q2: 19.72–21.22	0
	Q3: 21.22–22.60	0
	Q4: 22.60–24.77	1
	Q5: 24.77–29.05	1
PP	Q1: 25.00–35.00	−2
	Q2: 35.00–40.00	−1
	Q3: 40.00–50.00	0
	Q4: 50.00–55.00	1
	Q5: 55.00–80.00	2
FBG	Q1: 3.76–4.51	−2
	Q2: 4.51–4.93	−1
	Q3: 4.93–5.52	0
	Q4: 5.52–7.47	2
	Q5: 7.47–13.69	9
TG	Q1: 0.41–0.67	0
	Q2: 0.67–0.87	0
	Q3: 0.87–1.08	0
	Q4: 1.08–1.51	0
	Q5: 1.51–3.56	1
HDL-C	Q1: 0.47–0.78	5
	Q2: 0.78–0.96	3
	Q3: 0.96–1.18	2
	Q4: 1.18–1.45	0
	Q5: 1.45–2.04	−3

Abbreviations: MRS, metabolic risk score; BMI, body mass index; PP, pulse pressure; FBG, fasting blood glucose; TG, triglycerides; HDL-C, high-density lipoprotein cholesterol.

**Table 2 ijerph-19-15654-t002:** The demographics and pathological characteristics of patients in the training (*n* = 718) and the validation (*n* = 358) cohorts.

	Training Cohort (*n* = 718)	Validation Cohort (*n* = 358)
Age, mean ± SD	55.99 ± 9.54	56.42 ± 9.23
BMI, mean ± SD	26.20 ± 4.52	26.31 ± 4.46
SBP, mean ± SD	128.05 ± 16.04	128.48 ± 15.99
DBP, mean ± SD	78.75 ± 9.59	78.70 ± 9.80
PP, mean ± SD	49.30 ± 13.23	49.78 ± 12.68
FPG, mean ± SD	5.95 ± 1.66	5.94 ± 1.75
TC, mean ± SD	4.98 ± 1.05	4.94 ± 1.15
TG, mean ± SD	1.63 ± 0.86	1.56 ± 0.99
HDL-C, mean ± SD	1.24 ± 0.29	1.22 ± 0.34
MRS, mean ± SD	2.36 ± 4.31	2.58 ± 4.31
CA125, mean ± SD	52.41 ± 173.43	79.29 ± 395.40
CA199, mean ± SD	35.07 ± 85.80	44.73 ± 108.02
Menopause status, *n* (%)		
No	252 (35.10%)	119 (33.24%)
Yes	466 (64.90%)	239 (66.76%)
Diabetes, *n* (%)		
No	555 (77.3%)	287 (80.17%)
Yes	163 (22.7%)	71 (19.83%)
Hypertension, *n* (%)		
No	433 (60.31%)	202 (56.42%)
Yes	285 (39.69%)	156 (43.58%)
Ascites cytology		
Negative	665 (94.33%)	326 (92.88%)
Positive	40 (5.67%)	25 (7.12%)
LNM, *n* (%)		
Negative	661 (92.06%)	324 (90.50%)
Positive	57 (7.94%)	34 (9.50%)
Histology, *n* (%)		
Adenocarcinoma	659 (91.78%)	327 (91.34%)
others	59 (8.22%)	31 (8.66%)
Grade, *n* (%)		
G1	265 (36.91%)	133 (37.15%)
G2	324 (45.13%)	152 (42.46%)
G3	129 (17.97%)	73 (20.39%)
MI, *n* (%)		
<1/2	549 (76.46%)	278 (77.65%)
≥1/2	169 (23.54%)	80 (22.35%)
CI, *n* (%)		
Negative	649 (90.39%)	315 (87.99%)
Positive	69 (9.61%)	43 (12.01%)
LVSI		
Negative	597 (83.15%)	298 (83.24%)
Positive	121 (16.85%)	60 (16.76%)

Abbreviations: BMI, body mass index; SD, mean ± standard deviation; SBP, systolic blood pressure; DBP, diastolic blood pressure; PP, pulse pressure; FBG, fasting blood glucose; TC, total cholesterol; TG, triglycerides; HDL-C, high-density lipoprotein cholesterol; MRS, metabolic risk score; CA125, cancer antigen 125; CA199, cancer antigen 199; LNM, lymph node metastasis; MI, myometrial invasion; CI, cervical stromal invasion; LVSI, lymphovascular space invasion.

**Table 3 ijerph-19-15654-t003:** Univariate and multivariate analysis of LVSI in the training cohort.

	Univariate Analysis		Multivariate Analysis	
	OR (95% CI)	*p* Value	OR (95% CI)	*p* Value
Age	1.04 (1.02, 1.06)	0.0002	0.650 (0.379–1.116)	0.118
BMI	0.96 (0.91, 1.00)	0.0582		
MRS	1.15 (1.09, 1.20)	<0.0001	1.124 (1.067–1.185)	<0.001
SBP	1.00 (0.99, 1.01)	0.8016		
DBP	0.98 (0.96, 1.00)	0.1221		
PP	1.006 (0.99, 1.02)	0.414		
FPG	1.07 (0.96, 1.19)	0.2298		
TC	1.17 (0.97, 1.41)	0.1007		
TG	0.86 (0.67, 1.10)	0.2213		
HDL-C	0.59 (0.29, 1.18)	0.1340		
CA125	1.00 (1.00, 1.00)	0.0192	1.001 (1.000–1.002)	0.188
CA199	1.00 (1.00, 1.00)	0.5715		
Diabetes				
No	1.0			
Yes	1.42 (0.91, 2.21)	0.1214		
Hypertension				
No	1.0			
Yes	0.88 (0.59, 1.32)	0.5372		
Menopause status				
No	1.0			
Yes	2.11 (1.33, 3.33)	0.0015	1.617 (0.908–2.879)	0.103
Histology				
Adenocarcinoma	1.0			
Others	4.31 (2.46, 7.55)	<0.0001	2.723 (1.370–5.415)	0.004
Grade				
G1	1.0			
G2	2.15 (1.28, 3.61)	0.0037	1.800 (1.050–3.07.)	0.032
G3	5.26 (3.00, 9.24)	<0.0001	3.490 (1.870–6.520)	<0.001
MI				
<1/2	1.0			
≥1/2	5.26 (3.48, 7.96)	<0.0001	4.286 (2.663–6.896)	<0.001

Abbreviations: LVSI, lymphovascular space invasion; BMI, body mass index; MRS, metabolic risk score; SBP, systolic blood pressure; DBP, diastolic blood pressure; PP, pulse pressure; FBG, fasting blood glucose; TC, total cholesterol; TG, triglycerides; HDL-C, high-density lipoprotein cholesterol; CA125, cancer antigen 125; CA199, cancer antigen 199; MI, myometrial invasion.

**Table 4 ijerph-19-15654-t004:** Univariate and multivariate analysis of LVSI in the validation cohort.

	Univariate Analysis		Multivariate Analysis	
	OR (95% CI)	*p* Value	OR (95% CI)	*p* Value
Age	1.06 (1.02, 1.09)	0.0009	0.781 (0.505–1.208)	0.267
BMI	0.96 (0.90, 1.03)	0.2835		
MRS	1.09 (1.02, 1.16)	0.0083	1.092 (1.047–1.139)	<0.001
SBP	1.00 (0.98, 1.02)	0.8885		
DBP	0.99 (0.96, 1.02)	0.4505		
PP	1.00 (0.98, 1.03)	0.684		
FPG	1.11 (0.96, 1.27)	0.1564		
TC	1.06 (0.83, 1.35)	0.6330		
TG	0.83 (0.56, 1.22)	0.3372		
HDL-C	0.74 (0.31, 1.74)	0.4844		
CA125	1.00 (1.00, 1.00)	0.0990	1.000 (1.000–1.001)	0.156
CA199	1.00 (1.00, 1.00)	0.4396		
Diabetes				
No	1.0			
Yes	1.79 (0.95, 3.37)	0.0730		
Hypertension				
No	1.0			
Yes	0.99 (0.56, 1.73)	0.9669		
Menopause status				
No	1.0			
Yes	2.24 (1.14, 4.40)	0.0192	1.433 (0.885–2.322)	0.144
Histology				
Adenocarcinoma	1.0			
Others	5.87 (2.72, 12.71)	<0.0001	3.080 (1.778–5.338)	<0.001
Grade				
G1	1.0			
G2	1.98 (0.92, 4.23)	0.0787	1.880 (0.870–4.070)	0.110
G3	6.14 (2.81, 13.40)	<0.0001	3.650 (1.540–8.630)	0.003
MI				
<1/2	1.0			
≥1/2	4.96 (2.75, 8.95)	<0.0001	4.013 (2.720–5.920)	<0.001

Abbreviations: LVSI, lymphovascular space invasion; BMI, body mass index; MRS, metabolic risk score; SBP, systolic blood pressure; DBP, diastolic blood pressure; PP, pulse pressure; FBG, fasting blood glucose; TC, total cholesterol; TG, triglycerides; HDL-C, high-density lipoprotein cholesterol; CA125, cancer antigen 125; CA199, cancer antigen 199; MI, myometrial invasion.

## Data Availability

Not applicable.

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
