# Peer review of "Development and Validation of a Nomogram Based on Metabolic Risk Score for Assessing Lymphovascular Space Invasion in Patients with Endometrial Cancer"

_ijerph, 2022, doi:10.3390/ijerph192315654_

Round 1
Reviewer 1 Report
General comments:
The authors have assessed the predicative value of the metabolic risk score (MRS) for lymphovascular space invasion (LVSI) in 1076 endometrial cancer (EC) patients. An internal validation of the proposed model was performed through dividing the data into training and validation sets.
Overall, this speculative analysis exercise is well-written, with clear aims and an excellent introductory section. The supporting motivation for the manuscript is clearly outlined. The subject discussed is highly topical and covers an essential subject for the oncology community.
Specific comments:
1) Please state the limitations of the study in the abstract briefly. As acknowledged by the authors themselves in the discussion section, lines 324-337, " However, our study faced several limitations..." Nonetheless, many readers have access to the abstract and not always read the whole paper, and it is extremely important to avoid sending a wrong message.
2) The study luck of any external validation procedure (please add this to the limitation paragraph).
3) Please correct spelling mistakes on line 135.
Author Response
Dear reviewer,
Thank you very much for your reviewing our manuscript entitled “Development and validation of a nomogram based on metabolic risk score for assessing lymphovascular space invasion in patients with endometrial cancer”. We highly value your comments on our manuscript. Please see below our responses to your comments.
Point 1: Please state the limitations of the study in the abstract briefly. As acknowledged by the authors themselves in the discussion section, lines 324-337, " However, our study faced several limitations..." Nonetheless, many readers have access to the abstract and not always read the whole paper, and it is extremely important to avoid sending a wrong message.
Response 1: Thank you very much for your comments and we have added the limitations of our study in the abstract.
Point 2: The study luck of any external validation procedure (please add this to the limitation paragraph).
Response 2: Thanks a lot for your suggestions. We have added the lack of external validation in the limitation paragraph.
Point 3: Please correct spelling mistakes on line 135.
Response 3: Thank you for your comments. In fact, I failed to find spelling mistakes on line 135 due to the format change probably. But after careful examination, I found the spelling mistake of the word "addition" on line 247 in the first paragraph of the section of discussion. I wonder if it is what you mean.

Reviewer 2 Report
A. Introduction:
A-1 Please provide information about endometrial cancer epidemiology also in developing countries as is different from developed countries e.g. the United Sates. The following references might be helpful:
Banas T, Juszczyk G, Pitynski K, Nieweglowska D, Ludwin A, Czerw A. Incidence and mortality rates in breast, corpus uteri, and ovarian cancers in Poland (1980–2013): an analysis of population-based data in relation socio-economic changes. OncoTargets and Therapy 2016, 9:4121-4127.
Piechocki M, Koziołek W, Sroka D, Matrejek A, Miziołek P, Saiuk N, Sledzik M, Jaworska A, Bereza K, Pluta E, Banas T. Trends in Incidence and Mortality of Gynecological and Breast Cancers in Poland (1980-2018). Clin Epidemiol. 2022 Jan 24;14:95-114. doi: 10.2147/CLEP.S330081. PMID: 35115839; PMCID: PMC8800373.
B. Material and Methods:
B-1. Please provide detailed information how Metabolic Rating Score was calculated and haw the cut-off values were defined.
B-2. Please provide detailed information what tool was used for patients random assessment to analysed groups
B-3. Metabolic syndrome is present in patients with endometrial adenocarcinoma while endometrioid, clear-cell, anaplastic endometrial cancer present different clinical phenotype (elderly women, lean women) is the normogram also effective in this kind of patients?
B-4 Critical issue: Authors claims that: “developed and validated a nomogram for predicting LVSI in patients with EC” however this nomogram includes data available from histopathological report after surgical treatment ie. Lymph node status – if surgery is done prediction of LVSI is pointless as we have definite histopathological information on LVSI. This normogram could be useful in patients disqualified from surgery or requiring fertility sparing treatment – in these cases however we do not have histopathological confirmed information concerning lymph node status. Additionally if LN involvement is present or suspected in imaging enforced therapy is required regardless of LVSI status. Therefore a question rise to develop normogram that omits LN status.
C. Tables.
C-1 All the abbreviations used in tables should be explained in footnotes.
Author Response
Dear reviewer,
Thank you very much for your reviewing our manuscript entitled “Development and validation of a nomogram based on meta-bolic risk score for assessing lymphovascular space invasion in patients with endometrial cancer”. We highly value your comments on our manuscript. Please see below our responses to your comments.
Point A-1: Please provide information about endometrial cancer epidemiology also in developing countries as is different from developed countries e.g. the United Sates.
Response 1: Thank you very much for your comments and we have added information about epidemiology in developing countries.
Point B-1: Please provide detailed information how Metabolic Rating Score was calculated and haw the cut-off values were defined.
Response B-1: Thanks a lot for your suggestions. We adopted the MRS constructed by the Institute of Clinical Medical Sciences, China-Japan Friendship Hospital in Beijing, China when all five metabolism-related fac-tors were analyzed in quintiles, and the detailed process is shown in Table 1.
Point B-2: Please provide detailed information what tool was used for patients random assessment to analysed groups
Response B-2: Thank you for your comments. Referring your comments, we realized some improper expressions in our manuscript. After enrollment, all patients were randomly divided into the training and vali-dation cohort in a ratio of 2:1 using random number method in SPSS statistical software (version 24.0).
Point B-3: Metabolic syndrome is present in patients with endometrial adenocarcinoma while endometrioid, clear-cell, anaplastic endometrial cancer present different clinical phenotype (elderly women, lean women) is the normogram also effective in this kind of patients?
Response B-3: Thank you very much for your kindly instruction. We agree with your opinion and appreciate your comment. Hierarchical analysis was carried out according to pathological types, which were divided into endometrioid carcinoma group and nonendometrioid carcinoma group. The Logistic regression was used to compare the impact of MRS on LVSI, and the P value was less than 0.05 in both groups. (The details are presented in the table below).
|
Pathology |
Number of cases |
OR (95% CI) |
P value |
|
Endometrioid carcinoma |
659 |
1.13 (1.08, 1.19) |
<0.0001 |
|
Nonendometrioid carcinoma |
59 |
1.20 (1.04, 1.37) |
0.0110 |
Point B-4: Critical issue: Authors claims that: “developed and validated a nomogram for predicting LVSI in patients with EC” however this nomogram includes data available from histopathological report after surgical treatment ie. Lymph node status – if surgery is done prediction of LVSI is pointless as we have definite histopathological information on LVSI. This nomogram could be useful in patients disqualified from surgery or requiring fertility sparing treatment – in these cases however we do not have histopathological confirmed information concerning lymph node status. Additionally if LN involvement is present or suspected in imaging enforced therapy is required regardless of LVSI status. Therefore, a question rises to develop nomogram that omits LN status.
Response B-4: Thank you very much for your kind comments. We agree with you and have modified the methods section in details by removing postsurgical parameters in univariable logistic regression analysis, including ascites cytology, lymph node metastasis, and cervical invasion, and adding CA125 and CA199 to the analysis. Then, we reconstructed the nomogram and validation it.
Point C-1: All the abbreviations used in tables should be explained in footnotes.
ResponseC-1: Thank you very much for your kind suggestions. We have added all the abbreviations of tables in footnotes in the manuscript.

Round 2
Reviewer 2 Report
All remarks were addressed correctly. However it should be clearly stated that myometrial infiltration (MI) forming part of the PRE-operative normogram was evaluated using imaging techniques (eg. US; MRI) not taken from the final histopathological report.
Author Response
Dear reviewer,
Thank you very much for your reviewing our manuscript entitled “Development and validation of a nomogram based on metabolic risk score for assessing lymphovascular space invasion in patients with endometrial cancer”. We highly value your comments on our manuscript. Please see below our responses to your comments.
Point 1: It should be clearly stated that myometrial infiltration (MI) forming part of the PRE-operative normogram was evaluated using imaging techniques (eg. US; MRI) not taken from the final histopathological report.
Response 1: Thank you very much for your kindly instruction. We agree with your opinion and appreciate your comment. And we have added relevant information on line 198.
